# Development and validation of the Japanese version of the Lesbian, Gay, Bisexual, and Transgender Development of Clinical Skills Scale

Yusuke Kanakubo[1,2,3]*, Yoshifumi Sugiyama[1,4], Eriko Yoshida[1,3,5], Takuya Aoki[1], Rieko Mutai[6], Masato Matsushima[1], Tadao Okada[2]

1 Division of Clinical Epidemiology, Research Center for Medical Sciences, The Jikei University School of Medicine, Minato-ku, Tokyo, Japan, 2 Kameda Family Clinic Tateyama, Tateyama, Chiba, Japan, 3 Nijiiro Doctors, Minato-ku, Tokyo, Japan, 4 Division of Community Health and Primary Care, Center for Medical Education, The Jikei University School of Medicine, Minato-ku, Tokyo, Japan, 5 Department of General Internal Medicine, Kawasaki Kyodo Hospital, Kawasaki Health Cooperative Association, Kawasaki, Kanagawa, Japan, 6 Department of Adult Nursing, The Jikei University School of Nursing, Chofu, Tokyo, Japan

* kanakubo.research@gmail.com

**Data Availability Statement:** Data cannot be shared publicly because they contain potentially sensitive information. For example, it may be

## Abstract

### Introduction

The Lesbian, Gay, Bisexual, and Transgender Development of Clinical Skills Scale (LGBT-DOCSS) is a validated self-assessment tool for health and mental health professionals who provide healthcare for sexual and gender minority patients. This study aimed to develop and validate a Japanese version of LGBT-DOCSS (LGBT-DOCSS-JP) and examine its psychometric properties.

### Methods

LGBT-DOCSS was translated into Japanese and cross-culturally validated using cognitive debriefing. We then evaluated the structural validity, convergent and discriminant validity, internal consistency, and test–retest reliability of LGBT-DOCSS-JP using an online survey.

### Results

Data were analyzed for 381 health and mental health professionals aged 20 years or older from three suburban medical institutions. The confirmatory factor analysis indicated that the original three-factor model did not fit well with LGBT-DOCSS-JP. Exploratory factor analysis revealed four new factors: Attitudinal Awareness, Basic Knowledge, Clinical Preparedness, and Clinical Training. Convergent and discriminant validity were supported using four established scales that measured attitudes toward lesbians and gay men, genderism and transphobia, authoritarianism and conventionalism, and social desirability. The internal consistency of LGBT-DOCSS-JP was supported by the Cronbach's alpha values for the overall scale (0.84), and for each of its subscales (Attitudinal Awareness and Basic

possible for the staff of the medical institutions involved in this study to identify individual participants and their sexualities by combining the information contained in the data. This is ethically problematic because sexual and gender minority populations are at high risk of discrimination and prejudice in Japan. The ethics committees have imposed restrictions. Data are available from Division of Clinical Epidemiology, Research Center for Medical Sciences, The Jikei University School of Medicine (contact via clinicalepi@jikei.ac.jp) for researchers who meet the criteria for access to confidential data.

**Funding:** YK received funding from The Jikei University Research Fund for Graduate Students (grant number: N/A) (http://drclass.jikei.ac.jp/assistance/). The funders had no role in study design, data collection and analysis, decision to publish, or preparation of the manuscript.

**Competing interests:** I have read the journal's policy and the authors of this manuscript have the following competing interests: YK and EY serve as members of the board of Nijiiro Doctors, an organization dedicated to promoting awareness and education within the medical community regarding LGBTQ issues, while also providing support for the LGBTQ community. EY has received speaker's honoraria from Stryker Japan K. K. MM's son-in-law worked at IQVIA Services Japan K.K., which is a contract research organization and a contract sales organization. MM's son-in-law works at Syneos Health Clinical K. K. which is a contract research organization and a contract sales organization. The other authors declare that no competing interests exist.

Knowledge both 0.87, Clinical Preparedness 0.78, and Clinical Training 0.97). The test–retest reliability for the overall LGBT-DOCSS-JP was supported by an intraclass correlation coefficient score of 0.86.

## Conclusions

LGBT-DOCSS-JP has the potential to serve as a valuable tool in the development and assessment of effective curricula for LGBT healthcare education, as well as a means to promote self-reflection among trainees and professionals.

## Introduction

Sexual and gender minorities including lesbian, gay, bisexual, and transgender (LGBT) people have been reported to experience health and healthcare disparities [1, 2]. *Healthy People 2030*, the USA government objectives for improving health over the next decade, committed to eliminating LGBT health disparities [3]. It also emphasized the importance of enhancing efforts to improve LGBT health including providing medical students with training to increase the provision of culturally competent healthcare [3]. Appropriate education and competency assessment are key to improving healthcare for LGBT people. A systematic review has shown that education improves the knowledge and attitudes of medical staff and students toward LGBT patients [4].

The Lesbian, Gay, Bisexual, and Transgender Development of Clinical Skills Scale (LGBT-DOCSS), developed by Bidell in 2017, is an important instrument for enhancing healthcare provided to sexual and gender minorities. This scale assesses the knowledge, attitudes, and skills of healthcare and mental healthcare professionals towards caring for LGBT patients [5]. The scale was created to develop competent clinical services for LGBT people and improve on the shortcomings of the previous scales, such as insufficient focus on transgender people, and not ensuring validity in a multinational context [5]. LGBT-DOCSS has been used in recent studies to show that sufficient healthcare education and experience improves medical students' clinical skills [6] and to develop guidelines and recommendations for the patient care of sexual and gender minorities [7].

In Japan, health and healthcare disparities exist for LGBT people. LGBT people have higher rates of psychological distress, suicidal ideation, and suicide attempts than cisgender heterosexuals [2]. Although it is illegal to reject a patient on the basis of sexual orientation or gender identity, access to healthcare is limited and problematic, with about 50% of transgender people having had an unpleasant experience when visiting a medical institution, and about 50% hesitating to visit a medical institution when unwell [8]. Gender-affirming hormone therapy is not covered by national health insurance, and as of April 2023, only eight medical institutions have been authorized to perform gender-affirmation surgery by national health insurance. An anonymous survey of more than 8,000 physicians in Japan revealed that 46% had not seen any LGBT patients in the past 5 years [9]. In a study of nursing directors, more than 30% of hospitals limited ICU visitation and end-of-life care attendance to relatives and opposite-sex partners and did not allow same-sex partners [10]. These problems may partially reflect healthcare professionals' lack of knowledge and experience with LGBT health issues.

The provision of education on the healthcare of sexual and gender minorities is inadequate in Japan. For the first time, the 2016 revision of the model core curriculum for medical education emphasized the importance of explaining sexual orientation and gender identity as a core

competency [11]. A study in the USA recommended providing 35 hours of education on LGBT issues to medical students [6]. However, a recent survey showed that medical schools in Japan spend a median of 1 hour (interquartile range (IQR) 0–2 hours) compared with 4 (IQR 2–6) hours in the USA and Canada [12]. The model core curriculum for nursing education also sets understanding and assessing sexual diversity as a learning objective [13]. However, in a previous study, approximately 80% of nurses indicated that they did not learn about sexual and gender minorities in their basic nursing education [14]. One major obstacle is the lack of a suitable LGBT healthcare competence evaluation scale. Bidell concluded that the LGBT-DOCSS could be used to facilitate trainees' and healthcare professionals' self-exploration of their LGBT clinical skills to improve the areas of weakness [5]. However, the original LGBT-DOCSS could not be used directly in Japanese clinical settings because we needed a Japanese scale validated with a population of diverse healthcare professionals: the original LGBT-DOCSS was an English scale and most of the participants in Bidell's study were psychologists or psychology students [5]. The development of a Japanese tool for evaluating the clinical skills required to provide LGBT healthcare is urgently needed to improve the capacity of healthcare students and staff, and overcome the health disparities faced by sexual and gender minorities in Japan.

We therefore aimed to develop a Japanese version of LGBT-DOCSS (LGBT-DOCSS-JP) and examine the scale's reliability and validity among healthcare and mental healthcare professionals in Japan.

## Methods

This study took place in two phases. The first was translation of LGBT-DOCSS into Japanese, with cross-cultural adaptation in line with guidelines from the International Society for Pharmacoeconomics and Outcomes Research Task Force for Translation (ISPOR) [15]. We used standard Japanese, the version of the Japanese language that is widely used across the country and appropriate for almost all Japanese healthcare professionals to use and understand. The second was evaluation of reliability and validity of LGBT-DOCSS-JP using an online survey in line with the COnsensus-based Standards for the selection of health Measurement INstruments (COSMIN) guidelines [16]. We recruited healthcare and mental healthcare professionals from Kameda General Hospital (tertiary teaching hospital), Kameda Clinic (multispecialty outpatient clinic), and Kameda Family Clinic Tateyama (family medicine clinic) as research participants. All these locations are in suburban areas within a 10-km radius of Chiba, Japan. Ethical approval was obtained from the Ethics Committees of both The Jikei University School of Medicine (approval numbers: 33-364(10988) and 34-134(11285)) and Kameda Medical Center (approval numbers: 21–012 and 22–019).

### First phase: Development of LGBT-DOCSS-JP

We developed LGBT-DOCSS-JP in line with ISPOR guidelines [15]. With the original author's permission, three authors (YK, EY, YS), all of whom were native Japanese speakers and two of whom were familiar with sexual and gender minority issues, translated the original LGBT-DOCSS into Japanese. All the authors discussed cross-cultural adaptation and made a provisional version. Next, an English-Japanese bilingual physician, who had sufficient experience in back-translation [17–19] and was not familiar with LGBT-DOCSS, back-translated the provisional version into English. The original author reviewed the back-translated version and checked the discrepancies. After completing a prototype version, we recruited participants who were native Japanese speakers via personal connections by purposive sampling to conduct a cognitive debriefing. The ISPOR guidelines suggest that the newly translated measure should

be tested for cognitive equivalence by five to eight respondents who are native speakers of the target language [15]. In recruiting participants, we considered age, sexual orientation, gender identity, occupation, and years of professional experience to ensure maximum variation. Written informed consent was obtained from all participants. The primary investigator (YK) asked the participants to fill out the prototype version questionnaire and conducted face-to-face interviews individually using an interview guide to check interpretation, understandability, cultural adaptation, and alternative wording to make the scale better. All interviews were recorded and transcribed. Interviews were conducted between January 28[th] and March 9[th], 2022. All the authors discussed the results, and revised the prototype version before it was reviewed again by the original author, and a final version of LGBT-DOCSS-JP developed. Content validity and cross-cultural validity were verified through these processes.

## Second phase: Evaluation of reliability and validity of LGBT-DOCSS-JP

**Study design and data collection.**    We conducted an online survey among health and mental health professionals aged 20 years or older including physicians (including dentists), nurses, pharmacists, and psychologists affiliated with the study facilities. Two types of mass emails containing a link to self-administered questionnaires were sent separately to all participants. The second set were used to examine the test–retest reliability. We asked participants to complete the two questionnaires voluntarily at intervals of 2 to 4 weeks. Before participation, participants were provided with study information and required to indicate their willingness to participate by opting in. The first survey was administered on July 11[th], 2022. Participants were asked to provide demographic information, and complete LGBT-DOCSS-JP and four additional scales (see "Measures" section) to examine structural validity, hypothesis testing, and internal consistency. We also included a Directed Question Scale ("Check option 6 here"), which is commonly used to detect "satisficing" or answering behaviors in which participants do not devote appropriate attentional resources to the survey [20]. The second survey was administered on July 28[th], 2022, and aimed to evaluate the test–retest reliability of LGBT-DOCSS-JP. Both surveys were anonymous, but participants were asked to set a password at the first survey and enter the same password at the second survey to enable us to connect their responses. Those who responded incorrectly to the Directed Question Scale in the first survey or entered invalid passwords in the second survey were excluded from the analysis. Respondents got no reward. Reminder emails were sent twice per survey to mitigate selection bias. The questionnaires were designed using the required-response function to ensure no item response omissions except items related to participants' sexuality such as sex assigned at birth, gender identity, and sexual orientation. The data were collected between July 11[th] and August 31[st], 2022.

## Measures

**Participants' characteristics.**    The demographic questionnaire asked about age groups, sex assigned at birth, gender identity, sexual orientation, and professional healthcare specialization. It also asked whether respondents were aware of any coworkers who identified as homosexual or transgender/transsexual, in both their present and previous workplaces (referred to as "homosexual coworker" or "trans coworker"), and whether they had close friends, relatives, or family members who identified as homosexual or transgender/transsexual (referred to as "homosexual friends/family" or "trans friends/family") using a four-point Likert-type scale (*yes*, *probably*, *probably not*, or *no*) [21]. There is a lack of consensus about the appropriate method for inquiring about sexual orientation and gender identity in Japan. Thus we used a validated approach from a previous population-based study [22].

**LGBT-DOCSS-JP.** The original LGBT-DOCSS consists of 18 items across three factors: "Basic Knowledge" (four items: 1, 2, 6, and 8), "Attitudinal Awareness" (seven items: 3, 5, 7, 9, 12, 17, and 18), and "Clinical Preparedness" (seven items: 4, 10, 11, 13, 14, 15, and 16). Each item was rated on a seven-point Likert-type scale using the anchors *strongly disagree* to *strongly agree* (1 = *strongly disagree*, 4 = *somewhat agree/disagree*, 7 = *strongly agree*). Eight of the 18 were inverted items: 3, 4, 5, 7, 9, 12, 17, and 18. A reverse-scoring Likert-type scale was used for the inverted items (i.e., seven points were given for choice 1). The total mean score was determined by adding all test items and dividing the result by 18. The mean subscale scores were calculated by summing the scores of the corresponding questions and dividing the sum by the number of questions in each subscale. The total mean score and the mean subscale scores are therefore standardized on a scale of 1 to 7 points, with a higher score indicating greater healthcare competency.

**The Japanese 6-item revised version of the Attitudes Toward Lesbians and Gay Men Scale (ATLG-J6R).** ATLG-J6R is a six-item instrument designed to evaluate the degree of condemnation or tolerance held by heterosexual laypeople toward lesbian and gay populations [23]. Both the original full scale, consisting of 20 items, and a revised version, consisting of 6 items, have been validated by Horikawa et al. [23]. ATLG-J6R uses a seven-point Likert-type scale and has two subscales: "Attitudes toward Lesbians" and "Attitudes toward Gay Men". It has adequate reliability and validity among cisgender-heterosexual college students in Japan [23]. We used all six items of ATLG-J6R. A higher score indicates a worse attitude. Although this scale was not used in the previous study by Bidell [5], it was important for us to verify construct validity by comparing the LGBT-DOCSS-JP with ATLG-J6R, which measures negative attitudes toward gay men and lesbians. We hypothesized that ATLG-J6R would be negatively correlated with the LGBT-DOCSS-JP, particularly with the attitudinal awareness subscale.

**The Japanese Version of the Genderism and Transphobia Scale-Revised Short Form (J-GTS-R-SF).** The Japanese version of the Genderism and Transphobia Scale-Revised (J-GTS-R) is a 22-item instrument to measure the prejudicial attitudes of laypeople toward transgender individuals [24]. Adequate reliability and structural validity have been demonstrated among college students in Japan [24]. J-GTS-R-SF, the abbreviated version of J-GTS-R containing 13 items, was also validated by Mori et al. [24]. It uses a seven-point Likert-type scale [24]. Both scales have the same two factors: "Genderism and Transphobia", and "Gender Bashing". We used all seven items of the "Genderism and Transphobia" subscale of J-GTS-R-SF, following Bidell's method [5]. A higher score shows a higher level of genderism and transphobia. In the previous paper by Bidell, GTS-R-SF, the original version of J-GTS-R-SF, was used and showed a strong and negative correlation with the attitudinal awareness subscale of LGBT-DOCSS (r = −0.84) [5]. We hypothesized that J-GTS-R-SF, which measures negative attitudes toward transgender, would be negatively correlated with the LGBT-DOCSS-JP, particularly with the attitudinal awareness subscale.

**Japanese versions of the Right-Wing Authoritarianism scale (J-RWA).** J-RWA is a 30-item instrument with two factors ("Authoritarianism" and "Conventionalism") and a method factor [25]. It has adequate reliability and validity among survey panel members in Japan [25]. We selected five items focusing on sexual morality, homosexuality, sexual preference, gays and lesbians, and premarital sexual intercourse, in line with Bidell's study [5]. J-RWA uses a nine-point Likert-type scale, ranging from −4 = very strongly disagree to 4 = very strongly agree, but we used the seven-point Likert-type scale from the LGBT-DOCSS-JP (1 = strongly disagree to 7 = strongly agree) to minimize participants' fatigue with multiple scale changes, following Bidell's method [5]. A higher score indicates a higher level of authoritarianism and conventionalism. In the previous paper by Bidell, the same five items of RWA, the original version of J-RWA, were used and showed a strong and negative correlation with the attitudinal

awareness subscale of LGBT-DOCSS (r = −0.62) [5]. We hypothesized that J-RWA, which measured negative attitudes toward transgender, would be negatively correlated with the LGBT-DOCSS-JP, particularly with the attitudinal awareness subscale.

**The Japanese version of Balanced Inventory of Desirable Responding (J-BIDR).** J-BIDR is a 24-item instrument that quantifies social desirability, and contains two distinct factors: "Self-Deceptive Enhancement" and "Impression Management" [26]. Its reliability and validity have been rigorously examined among a sample of college students in Japan, using a seven-point Likert-type scale [26]. The "Self-Deceptive Enhancement" factor assesses the extent to which respondents truthfully depict themselves as socially desirable, and the "Impression Management" factor evaluates the degree to which respondents misrepresent themselves to manipulate their self-presentation. Certain items within the LGBT-DOCSS-JP contain socially undesirable or negative expressions towards LGBT individuals, which may result in a bias toward positive responses. To investigate the correlation between the LGBT-DOCSS-JP and social desirability, all 24 items of the J-BIDR were included in the study. A higher score shows a greater inclination toward socially desirable responding. In the previous paper by Bidell, the Marlowe-Crowne Social Desirability-Short Form-A [27], another scale for measuring social desirability, was used and showed a weak and negative correlation with the total score of LGBT-DOCSS (r = −0.16) [5]. However, we used J-BIDR instead because it is commonly used in Japan because of its stable factor structure [26]. We hypothesized that J-BIDR, which measures social desirability, would not be correlated with the LGBT-DOCSS-JP.

**Sample size.** The COSMIN guidelines suggest that studies on the development of patient-reported outcome measures should include a minimum of 100 participants [16]. The optimal subjects-to-variables ratio for exploratory factor analysis ranges from 3:1 to 20:1 [28]. LGBT-DOCSS contains 18 items, and it was therefore decided to include a minimum of 360 participants to ensure robust results.

## Statistical analysis

Descriptive statistics of participants' characteristics were generated for the study population. We also examined the psychometric properties of the questionnaires against the COSMIN guidelines [16].

**Structural validity.** We carried out a confirmatory factor analysis with maximum likelihood estimation to assess the structural validity of the original three-factor model (Attitudinal Awareness, Basic Knowledge, and Clinical Preparedness) [5]. We used several criteria to evaluate the fit of the model, including the comparative fit index (CFI), Tucker–Lewis index (TLI), root mean square error of approximation (RMSEA), and standardized root mean square residual (SRMR). Generally, values close to 0.95 or higher for CFI and TLI, less than 0.06 for RMSEA, and less than 0.08 for SRMR are considered to indicate a good model fit [29]. The model fit was inadequate, and we therefore carried out an exploratory factor analysis using promax rotation with maximum likelihood estimation to ascertain the structure of LGBT-DOCSS-JP. The determination of the number of factors to retain was based on Cattell's scree test [30]. Only items with a factor loading exceeding 0.30 were incorporated.

**Hypothesis testing.** We compared LGBT-DOCSS-JP with the other outcome measurement instruments to evaluate convergent and discriminant validity. Convergent validity was assessed by examining Spearman's rank correlation coefficients between the scores on LGBT-DOCSS-JP, including its overall score and subscales, and the scores on ATLG-J6R, J-GTS-R-SF, and J-RWA. Previous studies suggested that the LGBT-DOCSS-JP Attitudinal Awareness subscale would have the strongest correlation with ATLG-J6R, J-GTS-R-SF, and J-RWA. Conversely, to support discriminant validity, we hypothesized that the scores on

LGBT-DOCSS-JP would demonstrate minimal correlation with J-BIDR. Certain overarching principles may be used when interpreting correlation coefficients of varying magnitude. For example, a coefficient of 0.10 is deemed small, a coefficient of 0.30 is considered medium, and a coefficient of 0.50 is deemed large [31]. Convergent validity is often considered adequate when the correlation with a measure evaluating the same construct exceeds 0.50 [32].

We also compared the scores of LGBT-DOCSS-JP among subgroups. Several studies have reported a correlation between being relatively young and affirmative attitudes toward LGBT individuals [2, 21]. We hypothesized that scores on the LGBT-DOCSS-JP scale would be higher among younger than older adults. To test this hypothesis, we used Jonckheere–Terpstra tests to compare scores among the five age groups.

Previous studies have shown that participants identifying as LGBT show significantly higher scores on the LGBT-DOCSS scale [5]. We hypothesized that scores would be higher among individuals classified as sexual and gender minorities compared with cisgender-heterosexual participants. To test this hypothesis, we used a Wilcoxon rank sum test to compare scores between cisgender-heterosexual participants and individuals classified as "sexual and gender minorities (LGBTQA)", which includes lesbian, gay, bisexual, transgender, queer/questioning (including those who have not decided their sexualities), and asexual. Data from individuals who were missing responses to the questions on sexuality or who did not understand the question were excluded from this analysis.

Some studies of intergroup contact theory have reported that contact with members of stigmatized groups can reduce prejudice and improve attitudes towards those groups. In particular, heterosexual individuals who report personal acquaintance with gay men or lesbians show significantly more favorable attitudes toward the gay community than those who do not have this contact [33, 34]. We therefore also hypothesized that participants who know more about sexual and gender minorities would have higher scores than those who are less aware. To test these hypotheses among cisgender-heterosexual participants, we used Jonckheere–Terpstra tests to compare scores among the four levels of awareness of individuals who identify as homosexual or transgender/transsexual in current and previous workplaces and among close friends, relatives, or family members.

**Internal consistency.** Cronbach's α was calculated to assess internal consistency of total and subscale scores. A score of 0.70–0.95 is considered an acceptable range [35].

**Test–retest reliability.** The temporal stability of the measure was evaluated by assessing its 2-to-4 week test–retest reliability using the intraclass correlation coefficient (ICC (2,1)). Values less than 0.5, between 0.5 and 0.75, between 0.75 and 0.9, and greater than 0.90 are indicative of poor, moderate, good, and excellent reliability [36]. Responses were determined to be from the same respondent when the passwords matched.

**Responsiveness and interpretability.** Responsiveness and interpretability of scores were not assessed.

All statistical analyses used R, version 4.2.1 (the R Foundation). For each analysis, we used a two-tailed significance level of $P < 0.05$. Missing values were limited to three items: sex assigned at birth, gender identity, and sexual orientations. We used listwise deletion for comparative analysis between LGBTQA and cisgender-heterosexual participants, and among cisgender-heterosexual individuals with varying levels of awareness about sexual and gender minorities. Missing values did not affect other analyses in which sexuality was not taken into account.

## Results

### First phase: Development of LGBT-DOCSS-JP

In the first phase, we recruited eight native Japanese-speaking participants, including three physicians (a family physician, a pediatrician, and a urologist), three nurses, a pharmacist, and

a psychologist. The participants' ages ranged from 27 to 54 years (median = 39, IQR = 30.5–47.5) and years of experience ranged from 3 to 32 (median = 9.5, IQR = 5–21.5). Four participants were cisgender males, and four were cisgender females including one lesbian. No transgender individuals were included. The median interview time (IQR) was 36 (27–38) minutes.

During the translation process, some expressions were adjusted to facilitate cultural adaptation. For example, some terms such as sexual orientation, gender identity, lesbian, gay, bisexual, transgender, cisgender, and the umbrella term LGBT are not widely known in Japan, and annotations were therefore added to provide clarity (S1 Appendix). Some participants mentioned that these annotations were useful.

The participants pointed out wording or expressions that required improvement, and we discussed the modifications later. For instance, most of the participants were not familiar with the concept of "institutional barriers". We therefore adopted the expression "systematic barriers, such as rules and customs" for semantic equivalence. The expression of choices in the original questionnaire (*strongly disagree–strongly agree*) was considered inadequate because of the variance in natural Japanese response patterns contingent on the item type. We therefore modified the expression of choices for some items inquiring about knowledge or experience (i.e., *not at all–very well* or *none–very much*) until we had a final version of LGBT-DOCSS-JP (S1 Appendix). Content validity and cross-cultural validity were established through these processes.

## Second phase: Evaluation of reliability and validity of LGBT-DOCSS-JP

A total of 414 (22.3%) out of 1855 participants responded to the first questionnaire (Fig 1). However, 33 of these responses were deemed incorrect from the Directed Question Scale [20], resulting in a total of 381 (20.5%) responses available for analysis. We obtained 89 (14.2%) suitable responses from physicians/dentists, 252 (22.6%) from nurses, 34 (31.8%) from pharmacists, and 6 (100%) from licensed or certified clinical psychologists. In total, 151 (36.5%) of the

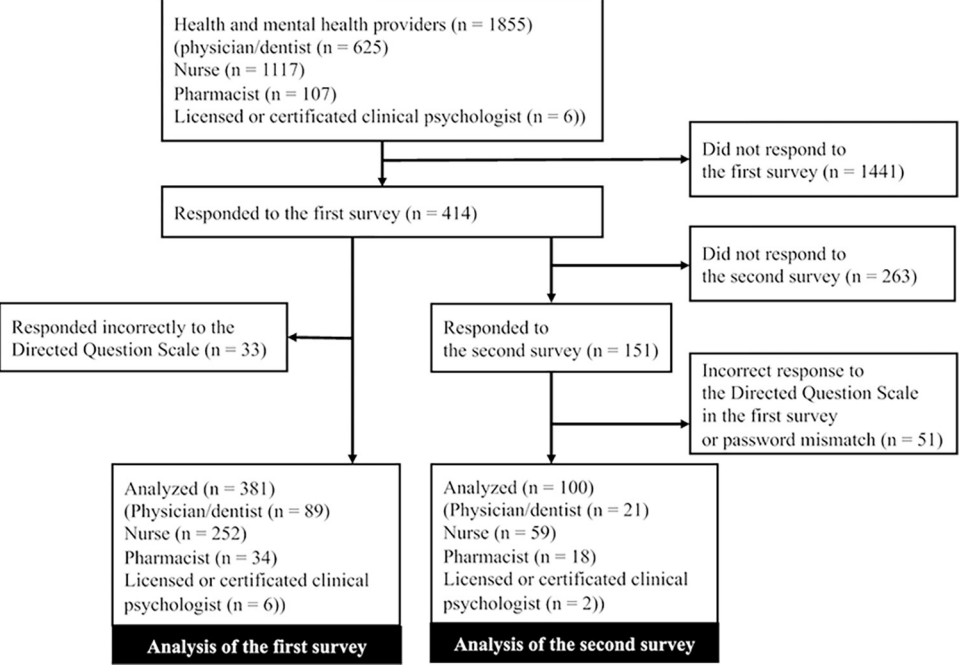

**Fig 1. Flow diagram of the study.**

**Table 1. Participants' characteristics.**

| | First survey | Second survey |
|---|---|---|
| | (n = 381) | (n = 100) |
| **Age (years), no. (%)** | | |
| 20–29 | 181 (47.5) | 35 (35.0) |
| 30–39 | 98 (25.7) | 34 (34.0) |
| 40–49 | 61 (16) | 21 (21.0) |
| 50–59 | 31 (8.1) | 7 (7.0) |
| $\geq$ 60 | 10 (2.6) | 3 (3.0) |
| **Sex assigned at birth, no. (%)** | | |
| Male | 108 (28.3) | 34 (34.0) |
| Female | 270 (70.9) | 65 (65.0) |
| Missing | 3 (0.8) | 1 (1.0) |
| **Gender identity, no. (%)** | | |
| Cisgender male | 106 (27.8) | 32 (32.0) |
| Cisgender female | 265 (69.6) | 63 (63.0) |
| Transgender | 7 (1.8) | 4 (4.0) |
| Missing | 3 (0.8) | 1 (1.0) |
| **Sexual orientation, no. (%)** | | |
| Heterosexual | 298 (78.2) | 74 (74.0) |
| Lesbian/gay | 4 (1.0) | 2 (2.0) |
| Bisexual | 11 (2.9) | 3 (3.0) |
| Asexual | 8 (2.1) | 3 (3.0) |
| Do not want to decide/have not decided | 34 (8.9) | 11 (11.0) |
| Do not understand the question | 8 (2.1) | 1 (1.0) |
| Other | 3 (0.8) | 2 (2.0) |
| Missing | 15 (3.9) | 4 (4.0) |
| **Professional healthcare specialization, no. (%)** | | |
| Physician (including dentist) | 89 (23.4) | 21 (21.0) |
| Nurse | 252 (66.1) | 59 (59.0) |
| Pharmacist | 34 (8.9) | 18 (18.0) |
| Licensed or certified clinical psychologist | 6 (1.6) | 2 (2.0) |

414 participants responded to the second questionnaire. However, 51 of these responses were excluded because of incorrect responses to the Directed Question Scale or password mismatch, giving a total of 100 responses for analysis.

Table 1 shows the baseline characteristics of participants. Overall, among 381 participants who were analyzed in the first survey, 47.5% (n = 181) of the sample were in their 20s, 69.6% (n = 265) identified as cisgender females, and 78.2% (n = 298) identified as heterosexual. The majority of participants (66.1%, n = 252) were nurses, and 87.3% (220 of 252) of the responding nurses were cisgender females. The mean (standard deviation (SD)) LGBT-DOCSS-JP total score was 4.16 (0.74) and the median (IQR) score was 4.11 (3.67–4.72). S1 Table shows the scores of cisgender participants and each professional healthcare specialization. The cisgender female participants' mean (SD) total score was 4.17 (0.68), and the nurses' mean (SD) total score was 4.13 (0.68).

**Structural validity.** A confirmatory factor analysis using the model described in the original paper [5] showed the following indices of model fit: CFI = 0.718, TLI = 0.673, RMSEA = 0.158 [90% confidence interval (CI) 0.151–0.166], and SRMR = 0.095. The model fit

**Table 2. Structure coefficients for LGBT-DOCSS-JP (n = 381).**

| | Item | Factor | | | |
|---|---|---|---|---|---|
| | | **1** | **2** | **3** | **4** |
| *9 | When it comes to transgender individuals, I believe they are morally deviant. | **0.829** | −0.042 | −0.086 | 0.018 |
| *17 | People who dress opposite to their biological sex have a perversion. | **0.808** | 0.083 | −0.056 | −0.049 |
| *12 | The lifestyle of a LGB individual is unnatural or immoral. | **0.796** | 0.010 | −0.071 | 0.032 |
| *18 | I would be morally uncomfortable working with a LGBT client/patient. | **0.713** | −0.001 | −0.020 | 0.035 |
| *3 | I think being transgender is a mental disorder. | **0.665** | −0.039 | 0.030 | 0.008 |
| *7 | LGB individuals must be discreet about their sexual orientation around children. | **0.555** | −0.008 | −0.002 | 0.045 |
| *5 | A same sex relationship between two men or two women is not as strong and as committed as one between a man and a woman. | **0.510** | −0.009 | 0.126 | −0.088 |
| 1 | I am aware of institutional barriers that may inhibit transgender people from using health care services. | −0.023 | **1.053** | −0.131 | −0.048 |
| 2 | I am aware of institutional barriers that may inhibit LGB people from using health services. | −0.014 | **1.022** | −0.112 | −0.033 |
| 8 | I am aware of research indicating that transgender individuals experience disproportionate levels of health and mental health problems compared to cisgender individuals. | 0.003 | **0.540** | 0.094 | 0.030 |
| 6 | I am aware of research indicating that LGB individuals experience disproportionate levels of health and mental health problems compared to heterosexual individuals. | 0.008 | **0.461** | 0.115 | 0.015 |
| 14 | I feel competent to assess a person who is LGB in a therapeutic setting. | 0.024 | −0.025 | **1.068** | −0.183 |
| 15 | I feel competent to assess a person who is transgender in a therapeutic setting. | 0.061 | −0.047 | **1.044** | −0.153 |
| 13 | I have experience working with LGB clients/patients. | −0.069 | 0.003 | **0.358** | 0.013 |
| 16 | I have experience working with transgender clients/patients. | −0.027 | −0.027 | **0.346** | 0.145 |
| *4 | I would feel unprepared talking with a LGBT client/patient about issues related to their sexual orientation or gender identity. | 0.087 | 0.066 | **0.310** | 0.130 |
| 11 | I have received adequate clinical training and supervision to work with lesbian, gay, and bisexual (LGB) clients/patients. | 0.009 | −0.004 | −0.043 | **1.015** |
| 10 | I have received adequate clinical training and supervision to work with transgender clients/patients. | −0.010 | −0.025 | −0.020 | **0.969** |
| | Proportion of variance | 0.19 | 0.14 | 0.13 | 0.10 |
| | Cronbach's α (subscale) | 0.87 | 0.87 | 0.78 | 0.97 |

Note. Extraction method: Maximum likelihood estimation. Rotation method: Promax rotation.

*Inverted item. Bold font shows factor loadings greater than 0.3.

was therefore deemed insufficient, and we carried out an exploratory factor analysis. The scree test suggested a four- or five-factor model. A five-factor model was rejected because it included items with low factor loading (< 0.30). However, the four-factor model was deemed interpretable, and all factor loadings exceeded 0.30. The factors were named "Attitudinal Awareness" (items 3, 5, 7, 9, 12, 17, and 18), "Basic Knowledge" (items 1, 2, 6, and 8), "Clinical Preparedness" (items 4, 13, 14, 15, and 16), and "Clinical Training" (items 10 and 11). Table 2 shows structure coefficients. The score distributions for the overall scale and each subscale are shown in S1 Fig.

**Hypothesis testing.** The results of hypothesis testing for convergent and discriminant validity are shown in Table 3. They showed adequate convergent and discriminant validity. The total scores for LGBT-DOCSS-JP showed moderate correlations with ATLG-J6R, J-GTS-R-SF, and J-RWA, indicating adequate convergent validity. The Attitudinal Awareness subscale showed the strongest correlation with the J-GTS-R-SF Genderism/Transphobia subscale (r = −0.70), followed by the ATLG-J6R (r = −0.63). The weak correlation between LGBT-DOCSS-JP and J-BIDR attests to the strong discriminant validity of LGBT-DOCSS-JP.

Table 4 shows the results of the hypothesis testing. The scores for the Attitudinal Awareness subscale were significantly higher among younger age groups. However, the scores for the Clinical Preparedness subscale were higher among older age groups. The overall scores, and the scores for the Basic Knowledge and Clinical Preparedness subscales, did not show significant differences. The cisgender-heterosexual participants had significantly lower scores for the

**Table 3. Convergent and divergent correlation matrix (n = 381).**

| | Attitudinal Awareness | Basic Knowledge | Clinical Preparedness | Clinical Training | ATLG-J6R | J-GTS-R-SF | J-RWA | J-BIDR |
|---|---|---|---|---|---|---|---|---|
| LGBT-DOCSS-JP Total scale scores | 0.54 | 0.74 | 0.72 | 0.51 | −0.47 | −0.50 | −0.40 | 0.18 |
| Attitudinal Awareness | | 0.13 | 0.20 | 0.06 | −0.63 | −0.70 | −0.50 | 0.14 |
| Basic Knowledge | | | 0.37 | 0.34 | −0.22 | −0.24 | −0.25 | 0.13 |
| Clinical Preparedness | | | | 0.44 | −0.20 | −0.22 | −0.16 | 0.12 |
| Clinical Training | | | | | −0.11 | −0.07 | −0.07 | 0.01 |
| ATLG-J6R | | | | | | 0.68 | 0.49 | −0.09 |
| J-GTS-R-SF | | | | | | | 0.52 | −0.18 |
| J-RWA | | | | | | | | −0.08 |

Note. Attitudinal Awareness, Basic Knowledge, Clinical Preparedness, and Clinical Training are subscales of LGBT-DOCSS-JP. ATLG-J6R = Japanese 6-item revised version of the Attitudes Toward Lesbians and Gay Men Scale; J-GTS-R-SF = items on the Genderism and Transphobia subscale of the Japanese Version of the Genderism and Transphobia Scale-Revised Short Form; J-RWA = items focusing on sexuality from the Japanese version of the Right-Wing Authoritarianism scale; J-BIDR = Japanese version of Balanced Inventory of Desirable Responding.

total scale and each subscale compared with LGBTQA participants. S2 Table shows that cisgender-heterosexual participants who were aware of homosexual coworkers, transgender/transsexual coworkers, and transgender/transsexual friends, family, or relatives had significantly higher overall scores and scores for some subscales. Those who were aware of sexual and gender minorities also had significantly higher scores on the Clinical Preparedness subscale in each subgroup analysis.

**Internal consistency.** In this group, Cronbach's alpha of LGBT-DOCSS-JP total score was 0.84 [95% CI 0.81–0.86]. The Cronbach's alpha for the four factors was 0.87 for both Attitudinal Awareness and Basic Knowledge, 0.78 for Clinical Preparedness, and 0.97 for Clinical Training.

**Table 4. Hypothesis testing (n = 381).**

| | No. | Total (mean (SD)) | Attitudinal Awareness (mean (SD)) | Basic Knowledge (mean (SD)) | Clinical Preparedness (mean (SD)) | Clinical Training (mean (SD)) |
|---|---|---|---|---|---|---|
| Total sample | 381 | 4.16 (0.74) | 6.30 (0.82) | 3.85 (1.51) | 2.37 (1.14) | 1.75 (1.21) |
| Age (years) | | | | | | |
| 20–29 | 181 | 4.13 (0.68) | 6.37 (0.82) | 3.85 (1.44) | 2.18 (1.04) | 1.73 (1.09) |
| 30–39 | 98 | 4.23 (0.82) | 6.39 (0.75) | 3.68 (1.77) | 2.60 (1.20) | 1.87 (1.52) |
| 40–49 | 61 | 4.06 (0.72) | 6.14 (0.88) | 3.88 (1.37) | 2.26 (1.08) | 1.58 (1.00) |
| 50–59 | 31 | 4.18 (0.76) | 5.97 (0.89) | 4.05 (1.27) | 2.74 (1.36) | 1.78 (1.13) |
| 60+ | 10 | 4.35 (0.88) | 6.10 (0.75) | 4.50 (1.30) | 2.80 (1.26) | 1.80 (1.23) |
| P value[a] | | 1 | **< 0.01** | 0.50 | **0.01** | 0.56 |
| Sexual orientation and gender identity | | | | | | |
| Cisgender-heterosexual | 293 | 4.09 (0.70) | 6.26 (0.85) | 3.70 (1.44) | 2.30 (1.14) | 1.72 (1.23) |
| Sexual and gender minorities (LGBTQA) | 62 | 4.49 (0.78) | 6.51 (0.63) | 4.50 (1.58) | 2.72 (1.12) | 1.85 (1.17) |
| P value[b] | | **< 0.01** | **< 0.01** | **< 0.01** | **< 0.01** | **< 0.01** |

Note. LGBT-DOCSS-JP total scale scores and subscale scores are described.

a) Jonckheere–Terpstra test.

b) Wilcoxon rank sum test. Bold font shows significance at P < 0.05 in P value line. SD, standard deviation.

**Test-retest reliability.** ICC(2,1) of LGBT-DOCSS-JP total score was 0.86 [95% CI 0.80–0.91]. The values for the four factors were 0.77 for Attitudinal Awareness, 0.80 for Basic Knowledge, 0.71 for Clinical Preparedness, and 0.82 for Clinical Training.

## Discussion

This paper describes the process of developing LGBT-DOCSS-JP through a cross-cultural validation process following a scale translation guideline and examining its psychometric properties. We confirmed the reliability and validity of LGBT-DOCSS-JP.

In terms of structural validity, LGBT-DOCSS-JP has a distinct four-factor structure, covering Attitudinal Awareness, Basic Knowledge, Clinical Preparedness, and Clinical Training. The Attitudinal Awareness and Basic Knowledge subscales include the same items as the original LGBT-DOCSS, but we separated the original scale's Clinical Preparedness subscale into two domains: Clinical Preparedness and Clinical Training. One of the main reasons is the difference in educational systems across nations. Healthcare education on LGBT issues is lacking within the USA, but the deficiency is even more conspicuous in Japan. A recent study reported that the proportion of medical schools that did not teach LGBT content at all during clinical training was significantly higher in Japan than in the USA and Canada (47.2% vs 33.3%) [12]. Approximately 80% of nurses indicated that they did not learn about sexual and gender minorities in their basic nursing education in Japanese nursing schools [14]–a much higher proportion than the 28% of baccalaureate nursing programs in the USA that did not teach LGBT sexual health at all [37]. In a survey of nursing directors of 252 hospitals in Japan, more than 90% indicated that they do not provide continuing nursing education on LGBT issues, and approximately 10% reported that healthcare education is not necessary [10]. These training issues in Japan are mainly due to the unavailability of suitable instructors and most schools lacking policies regarding LGBT healthcare education [12, 37, 38].

It should be noted that despite the lack of official practical training, many healthcare professionals may be aware of the needs of LGBT patients and provide healthcare for them regularly, increasing their clinical preparedness through trial and error. The discrepancy between the lack of official healthcare education and actual clinical practice may account for the separation of the original Clinical Preparedness factor into two factors in our study. This separation was also supported by comparing the scores with the original study [5]. The mean (SD) scores in our study for the Clinical Preparedness and Clinical Training subscales were 2.37 (1.14) and 1.75 (1.21), notably lower than the Clinical Preparedness score of 3.51 (1.45) reported in the original study [5]. The Basic Knowledge subscale mean (SD) score was also lower in our study than in the original study (3.85 (1.51) vs. 4.95 (1.51)). The low scores on these three subscales may be because of insufficient healthcare education in Japan. Providing precise knowledge and on-the-job clinical training with LGBT patients may be effective in the current Japanese context. However, the Attitudinal Awareness subscale score was as high in this study as in the original study (6.30 (0.82) vs. 6.52 (0.72)) [5]. These findings align with the social milieu of Japan. Japan is the only G7 country that has not introduced same-sex marriage or partnership that guarantees the same rights as marriage, at the national level. However, the 2010s saw a growing movement among citizens to recognize same-sex partnerships at the municipal level. Currently, more than 300 municipalities recognize same-sex partnerships, with a population coverage rate of more than 70%, and the proportion of citizens showing favorable views of LGBT individuals is increasing [21]. In contrast, government policy has not kept pace with public opinion and local government policy, and no legislation has yet been passed. A higher score for Attitudinal Awareness is therefore reasonable.

We also conducted some hypothesis testing. As hypothesized, younger people scored better in the Attitudinal Awareness subscale, which is in line with previous research [5, 21]. Older participants scored higher on the Clinical Preparedness subscale, probably because of increased clinical expertise acquired over their years of professional healthcare experience. This is inconsistent with a previous study of healthcare providers in the USA, which reported no significant difference in LGBT-DOCSS scores based on age [39]. Further research in other settings is required to confirm these results. As expected, LGBTQA participants scored higher on LGBT-DOCSS-JP than cisgender-heterosexual participants. Cisgender-heterosexual individuals who were aware of homosexual coworkers at their workplace or knew there were homosexual or transgender/transsexual individuals among their friends, relatives, or family scored higher than their peers. These findings are in line with previous studies [33, 34]. However, awareness of transgender/transsexual coworkers was not significantly associated with higher scores except for the Clinical Preparedness subscale. The literature offers inconsistent conclusions about the effectiveness of contact with transgender people in reducing bias. A critical shortcoming is the limited measurement of intergroup contact with transgender people [40]. Previous studies have used a dichotomous measure of contact, disregarding the importance of both quantity and quality of contact [40]. Our study asked solely about participants' perceptions of the presence or absence of transgender/transsexual individuals. Consequently, our measure may have failed to accurately capture the frequency and quality of intergroup contact, potentially undermining the observed effects of this contact on bias reduction. The restricted number of participants may also explain the lack of statistical power.

Our analyses showed satisfactory validity. LGBT-DOCSS-JP, especially the Attitudinal Awareness subscale, showed a robust association with established scales assessing attitudes toward lesbian and gay individuals, genderism and transphobia, as well as authoritarianism and conventionalism. It was not correlated with a social desirability scale unrelated to the construct of LGBT-DOCSS-JP. These findings show sufficient convergent and discriminant validity of LGBT-DOCSS-JP.

The results also suggest that the scale is reliable. The Cronbach's alpha and ICC(2,1) values of LGBT-DOCSS-JP total score were sufficiently good, and almost the same as those reported in the original study [5]. These findings indicate that LGBT-DOCSS-JP shows robust internal consistency and test–retest reliability.

This study has numerous strengths. First, LGBT-DOCSS-JP is the first Japanese scale to assess health and mental health professionals' clinical skills with LGBT patients. However, this scale was not designed to serve as a high-stakes assessment such as a pass–fail comprehensive degree examination, certification test, or licensing examination [5]. Second, the translation is based on a culturally and academically validated process. Third, we had a sufficiently large sample to evaluate psychometric properties, and the results of hypothesis testing were also conceptually acceptable. Fourth, we used an online survey to guarantee anonymity because of the sensitive nature of the topic. Some participants may not have engaged fully with the survey, but we included a question to detect and mitigate this drawback [20]. Fifth, the study demonstrated the applicability of LGBT-DOCSS in non-English-speaking and non-Western countries and cultures, addressing a limitation of the previous study [5].

This study also had some limitations. First, it was conducted exclusively within three suburban medical institutions, imposing constraints on its external validity. Future investigations are needed in diverse clinical settings. Second, there was some self-selection bias. Our respondents included a significantly higher percentage of sexual and gender minorities (LGBTQA) compared with a previous general population study (22.8% vs. 8.2%) [2]. Additionally, approximately four times as many individuals responded "*yes*" or "*probably*" about their awareness of sexual and gender minorities than in a previous study [21]. Those who had more awareness of

sexual and gender minorities may have been more inclined to respond. Furthermore, the overall response rate was limited despite our reminder emails. It is therefore conceivable that our scale scores were overestimated when compared with the general population. Third, interpretability and responsiveness were not evaluated, and these psychometric properties should be confirmed in future studies. Fourth, there was a bias of gender and professional healthcare specialization among the participants. In this study, the percentages of cisgender females and nurses were both close to 70%, with 87.3% of responding nurses being cisgender females. According to the national statistics in 2020, 92% of nurses in Japan are female [41]. Thus, the percentage of female nurses in this study is representative of Japan. However, it is clear that the overall population scores were heavily influenced by cisgender female and nurse participants. Therefore, caution should be exercised when making comparisons with populations that have different distributions of gender and specialty areas among healthcare professionals, as is the case when making international comparisons.

In conclusion, LGBT-DOCSS-JP could be relevant for assessing the clinical skills of Japanese healthcare professionals working with people of diverse sexuality and gender. Nevertheless, a more extensive investigation is needed to confirm these results. LGBT-DOCSS-JP has the potential to help with the development and assessment of effective curricula for healthcare education and training in providing healthcare for LGBT people. This is urgently needed in Japan, where effective training programs and methods have yet to be established. LGBT-DOCSS-JP may also be a valuable tool to promote self-reflection among trainees and professionals about their LGBT attitudinal awareness, basic knowledge, clinical preparedness, and clinical training. We hope this scale will primarily be used for self-exploration and competency development.

## Supporting information

**S1 Appendix. The Japanese version of the Lesbian, Gay, Bisexual, and Transgender Development of Clinical Skills Scale (LGBT-DOCSS-JP).**
(PDF)

**S1 File. The original LGBT-DOCSS(English).**
(DOCX)

**S1 Fig. The score distributions for the overall scale and each subscale.**
(PDF)

**S1 Table. The scores of cisgender participants and each professional healthcare specialization.**
(PDF)

**S2 Table. Hypothesis testing among cisgender-heterosexual participants.**
(PDF)

## Acknowledgments

The authors are grateful to Dr. Toshiaki Kameda and Ms. Yaeko Watanabe for supporting the data collection. We also thank Dr. Daichi Hayashi for the back-translation. We would like to take this opportunity to thank Dr. Mamiko Ukai and Dr. Ryota Takahashi for their collaboration and advice as Kameda Family Clinic Tateyama primary care research team. We also thank Melissa Leffler, MBA, and Michelle Pascoe, PhD, from Edanz (https://jp.edanz.com/ac) for editing a draft of this manuscript.

## Author Contributions

**Conceptualization:** Yusuke Kanakubo, Yoshifumi Sugiyama, Eriko Yoshida, Takuya Aoki, Rieko Mutai, Masato Matsushima, Tadao Okada.

**Data curation:** Yusuke Kanakubo.

**Formal analysis:** Yusuke Kanakubo.

**Funding acquisition:** Yusuke Kanakubo.

**Investigation:** Yusuke Kanakubo, Yoshifumi Sugiyama, Eriko Yoshida.

**Methodology:** Yusuke Kanakubo, Yoshifumi Sugiyama, Eriko Yoshida, Takuya Aoki, Rieko Mutai, Masato Matsushima, Tadao Okada.

**Project administration:** Yusuke Kanakubo.

**Resources:** Yusuke Kanakubo.

**Software:** Yusuke Kanakubo, Yoshifumi Sugiyama, Takuya Aoki.

**Supervision:** Masato Matsushima, Tadao Okada.

**Validation:** Yusuke Kanakubo, Yoshifumi Sugiyama.

**Visualization:** Yusuke Kanakubo.

**Writing – original draft:** Yusuke Kanakubo, Yoshifumi Sugiyama.

**Writing – review & editing:** Yusuke Kanakubo, Yoshifumi Sugiyama, Eriko Yoshida, Takuya Aoki, Rieko Mutai, Masato Matsushima, Tadao Okada.

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
