## [Decision Letter · Decision Letter 0]

10 Oct 2023

PONE-D-23-21890Development and validation of the Japanese version of the Lesbian, Gay, Bisexual, and Transgender Development of Clinical Skills ScalePLOS ONE

Dear Dr. Kanakubo,

Thank you for submitting your manuscript to PLOS ONE. After careful consideration, we feel that it has merit but does not fully meet PLOS ONE’s publication criteria as it currently stands. Therefore, we invite you to submit a revised version of the manuscript that addresses the points raised during the review process. This study presents an interesting and potentially relevant addition to the current body of literature in this field. There are some issues with the current presentation of the study and significantly more information is required to ensure that potential readers have a robust understanding of all aspects of the study. I would like to as the authors to consider the comments made by reviewer 2 in particular.

We look forward to receiving your revised manuscript.

Kind regards,

Daniel Demant, PhD, MPH, GradCertHEd, BAppSocSc

Academic Editor

PLOS ONE

2.Thank you for stating the following in the Competing Interests section: 

[I have read the journal's policy and the authors of this manuscript have the following competing interests: YK and EY serve as members of the board of Nijiiro Doctors, an organization dedicated to promoting awareness and education within the medical community regarding LGBTQ issues, while also providing support for the LGBTQ community. MM’s son-in-law worked at IQVIA Services Japan K.K., which is a contract research organization and a contract sales organization. MM’s son-in-law works at Syneos Health Clinical K.K. which is a contract research organization and a contract sales organization.

The other authors declare that no competing interests exist.]. 

Additional Editor Comments:

While the article mentions the sample size rationale based on COSMIN guidelines, it should also mention how the sample size was determined for each phase.

The study assesses various scales and measures, it's important to explain why specific scales were chosen and how they align with the research objectives rather than just describing the scales themselves.

There are minor grammatical issues. Proofreading is needed for smoother language.

Reviewers' comments:

Reviewer's Responses to Questions

**Comments to the Author**

1. Is the manuscript technically sound, and do the data support the conclusions?

Reviewer #1: Partly

Reviewer #2: Partly

2. Has the statistical analysis been performed appropriately and rigorously? 

Reviewer #1: Yes

Reviewer #2: Yes

3. Have the authors made all data underlying the findings in their manuscript fully available?

Reviewer #1: No

Reviewer #2: No

4. Is the manuscript presented in an intelligible fashion and written in standard English?

Reviewer #1: Yes

Reviewer #2: Yes

5. Review Comments to the Author

Reviewer #1: The proposed article meets the publication criteria of the journal in whole or in part.

Minor modifications should be made to make the research process more transparent and offer a more modest interpretation of the results to better reflect the true contribution.

Introduction: It would be interesting to further contextualize the Bidell scale to understand under what circumstances it was developed. This would also clarify why this scale is not complete or adequate for full use in Japan.

Measures: The same goes for the new items in the Japanese version of the scale. The author does not justify changes or additions. The changes are only named (eg. 24 items), but they are not located in relation to the old version. How and why should these scales be changed to better reflect the Japanese context? What explains the addition, removal or modification of the items that make up each of the scales?

Conclusions: Overall the article makes an interesting contribution to the field. However, as the author mentions, the results come from very specific clinics which are not necessarily representative of other care contexts. This limitation should therefore be reflected more clearly in the conclusions. It can only be concluded that the new version of the scale could be relevant for studying the abilities to work with sexually diverse people of Japanese professionals, but a larger investigation will need to confirm these results.

Reviewer #2: Overall, the article adds to the current body of research. The creation of the additional factor of Clinical training is very interesting and could be developed further. The researchers did a great job with their analysis and the tools they used to establish the validity and reliability. However, there are some concerns.

The first concern is the researchers did not utilize a professional interpreter to interpret the original English version of the survey. It appears from the article, the researchers used the feedback from the participants to modify words that may not translate well, as explained in lines 334-343. It is important to note, were the participants native Japanese speakers. Are there different versions of Japanese spoken based on geographic location in Japan? As a reviewer, I have concerns if this is enough to translate a document.

The second concern is the paper does not identify, discuss, or address the cultural experiences of being LGBT in Japan. What is that overall experience and the healthcare disparities associated with being LGBT. Do healthcare disparities exist in Japan among LGBT people? If so, which ones and what role does the healthcare professional play? In Japan, are healthcare providers allowed to turn a patient away? Are there limitations in the care the healthcare provider can provide. Needs some clarification and more in-depth backgrounds.

The rest of my review is comprised of recommendations. There is a lack of consistency of the terms throughout the paper, including ‘education’ and ‘care’. It is my recommendation that when mentioning education be specific to health care education. When mentioning care it should be specific to ‘healthcare’ or ‘patientcare’. For example line 50 and lines 70-72 should be clear that it is health care education. Line 72 … missing the word patient, not just caring for sexual and gender minorities but patient care. Line 80 Be specific- lack of a suitable LGBT healthcare competence evaluation scale.

Another recommendation is providing the readers with a background/introduction into the medical school. For example, Line 74- the term medical school is specified- is this medical school for medical doctors? Does this include nursing? Try to be more specific if possible. Since most participants were nurses their required education should be addressed and discussed.

I recommend on Line 358 when providing percentages, they should provide the n so the reader knows that number that percentage is representing.

Introduction

There is a more updated version to Healthy People, Healthy People 2030.

Are there guidelines on how much time should be spent? The introduction should address the culture of Japan towards LGBT patients.

LGBT-DOCSS does tell how to teach this content but rather identifies areas of weakness in the healthcare practitioners (line 79).

First Phase-

The scoring was explained, specifically the reverse scoring of the attitudinal scale. Why are they so high and Clinical training, clinical preparedness, and basic knowledge are all low, clinical preparedness is really low, especially when looking at age- wouldn’t age have higher scores? They don’t explain the scoring.

Results

Close to 70% were cisgender females, are the majority of healthcare workers female in Japan? Is this representative? What role/impact does most participants being nurse and females have? Does this impact the interpretation of the survey?

Discussion

The researchers state despite the lack of official practical training, many healthcare professionals are aware and provide care- did they collect data on how many of participants provide care to LGBT and how many hours of training they received?

Is there continued education or other less formal training and education settings offered? Sentences 431-436- who is responsible for the education of the nurses? In one sentence the authors are discussing school and the next sentence the authors are talking about continuing education. There needs to be clarity because you cannot compare the two to each other. For example, “line 431 …over 90% of hospitals in Japan do not provide nursing training with LGBT content, despite the demand for education… this is much worse than the 28% of baccalaureate nursing programs’- these two are not comparable. You state however, many healthcare professionals’ area aware of LGBT needs and treat them regularly- what is this based on? Did you measure how many patients the HCW sees?

Line 394 Why would the scores for clinical preparedness be higher among older age groups?

Line 436 What is meant by school policy? Is there a governing body over the requirements of education or is it done on a school to school/institutional to institutional basis?

450 What is meant by same-sex partnership schemes.

Line 450 ‘where a succession of same-sex partnership schemes have been instituted at the municipal level since 2015, and the proportion of citizens. What does this mean?

6. PLOS authors have the option to publish the peer review history of their article (what does this mean?). If published, this will include your full peer review and any attached files.

Reviewer #1: No

Reviewer #2: No

---

## [Author Response · Author response to Decision Letter 0]

8 Jan 2024

Comments from the Editor:

#1

While the article mentions the sample size rationale based on COSMIN guidelines, it should also mention how the sample size was determined for each phase.

Thank you for this comment. The first phase of the study was conducted in line with ISPOR guidelines, while the second phase followed COSMIN guidelines. Therefore, the phase 1 sample size was determined using ISPOR guidelines, and the phase 2 sample size was based on COSMIN guidelines. We now describe the determination of the sample size, as follows:

Page 5, line 123

The second was evaluation of reliability and validity of LGBT-DOCSS-JP using an online survey in line with the COnsensus-based Standards for the selection of health Measurement INstruments (COSMIN) guidelines [16].

Page 5, line 145

The ISPOR guidelines suggest that the newly translated measure should be tested for cognitive equivalence by five to eight respondents who are native speakers of the target language [15].

#2

The study assesses various scales and measures, it’s important to explain why specific scales were chosen and how they align with the research objectives rather than just describing the scales themselves.

Thank you for raising this point. Hypothesis testing of construct validity requires high correlations among related constructs and low correlations among unrelated constructs. In Bidell’s paper on the development and validation of LGBT-DOCSS, the English versions of J-GTS-R-SF (which measured negative attitudes toward transgender people) and J-RWA (which measured authoritarianism and conventionalism) were used to verify construct validity. ATLG-6R measures negative attitudes towards gay men and lesbians. Although the English version of ATLG-J6R was not used in the previous research, we considered it essential to verify the correlation between LGBT-DOCSS-JP and J-GTS-R-SF, and LGBT-DOCSS-JP and ATLG-6R. This is because people’s attitudes toward transgender people and their attitudes toward gay men and lesbians differ. In the previous paper, Bidell also used the Marlowe-Crowne Social Desirability-Short Form-A, which measures social desirability, to verify the discriminant validity of LGBT-DOCSS. We used J-BIDR instead because this scale is commonly used to measure social desirability in a Japanese research context. Thus, we have now added some further explanations to the text, as follows:

Page 7, line 224

Although this scale was not used in the previous study by Bidell [5], it was important for us to verify construct validity by comparing the LGBT-DOCSS-JP with ATLG-J6R, which measures negative attitudes toward gay men and lesbians. We hypothesized that ATLG-J6R would be negatively correlated with the LGBT-DOCSS-JP, particularly with the attitudinal awareness subscale.

Page 8, line 240

In the previous paper by Bidell, GTS-R-SF, the original version of J-GTS-R-SF, was used and showed a strong and negative correlation with the attitudinal awareness subscale of LGBT-DOCSS (r = −0.84) [5]. We hypothesized that J-GTS-R-SF, which measures negative attitudes toward transgender, would be negatively correlated with the LGBT-DOCSS-JP, particularly with the attitudinal awareness subscale.

Page 8, line 257

In the previous paper by Bidell, the same five items of RWA, the original version of J-RWA, were used and showed a strong and negative correlation with the attitudinal awareness subscale of LGBT-DOCSS (r = −0.62) [5]. We hypothesized that J-RWA, which measured negative attitudes toward transgender, would be negatively correlated with the LGBT-DOCSS-JP, particularly with the attitudinal awareness subscale.

Page 9, line 277

In the previous paper by Bidell, the Marlowe-Crowne Social Desirability-Short Form-A [27], another scale for measuring social desirability, was used and showed a weak and negative correlation with the total score of LGBT-DOCSS (r = −0.16) [5]. However, we used J-BIDR instead because it is commonly used in Japan because of its stable factor structure [26]. We hypothesized that J-BIDR, which measures social desirability, would not be correlated with the LGBT-DOCSS-JP.

#3

There are minor grammatical issues. Proofreading is needed for smoother language.

Thank you for your comment. We have re-submitted this manuscript to Edanz (https://jp.edanz.com/ac) for another round of proofreading.

Reviewers’ Comments:

Reviewer 1

#1

The proposed article meets the publication criteria of the journal in whole or in part.

Minor modifications should be made to make the research process more transparent and offer a more modest interpretation of the results to better reflect the true contribution.

#2

Introduction: It would be interesting to further contextualize the Bidell scale to understand under what circumstances it was developed. This would also clarify why this scale is not complete or adequate for full use in Japan.

Thank you for your comments. Although there are several scales to measure clinicians’ skills and attitudes regarding sexual and gender minority patients in the USA, they are limited to gay men or lesbians. There was a need to develop a more inclusive measure focusing on homosexuals and transgender people. Furthermore, the cross-cultural validity of existing scales was limited. Therefore, Bidell developed a comprehensive assessment scale of clinical skills with LGBT patients, including participants from the USA and UK. However, the original LGBT-DOCSS could not be used directly in Japanese clinical settings for two reasons. First, the official language of Japan is Japanese, and medical education and practice in the medical field are also conducted in Japanese. Thus, a Japanese language scale was necessary. Second, the participants in Bidell’s research were mainly from the psychology discipline (e.g., psychologists, psychology students, and counselors). We wanted to develop a useful scale to measure the skills of healthcare professionals from diverse disciplines. Therefore, we recruited physicians, nurses, pharmacists, and psychologists. Of course, as noted in the discussion section, further research is needed to determine whether LGBT-DOCSS-JP, which was developed and validated at a limited number of medical institutions, can be used throughout Japan. Thus, we revised the paper as follows:

Page 3, line 68

The scale was created to develop competent clinical services for LGBT people and improve on the shortcomings of previous scales, such as insufficient focus on transgender people, and not ensuring validity in a multinational context [5].

Page 4, line 104

However, the original LGBT-DOCSS could not be used directly in Japanese clinical settings because we needed a Japanese scale validated with a population of diverse healthcare professionals: the original LGBT-DOCSS was an English scale and most of the participants in Bidell’s study were psychologists or psychology students [5].

#3

Measures: The same goes for the new items in the Japanese version of the scale. The author does not justify changes or additions. The changes are only named (eg. 24 items), but they are not located in relation to the old version. How and why should these scales be changed to better reflect the Japanese context? What explains the addition, removal or modification of the items that make up each of the scales?

Thank you for your constructive comments around this point, which we have now clarified in the manuscript. The LGBT-DOCSS developed by Bidell in the USA was translated into Japanese in this study. No items were added or deleted. When translating the LGBT-DOCSS into Japanese, there were some concepts in the original English scale that do not exist in Japanese – a common problem when translating a tool from one language to another. Those concepts were paraphrased in the adapted scale so that they would be understood naturally in Japanese. The validity of this paraphrasing was confirmed in the first phase of the study. We have now described this process more clearly, as follows:

Page 12, line 390

The participants pointed out wording or expressions that required improvement, and we discussed the modifications later. For instance, most of the participants were not familiar with the concept of “institutional barriers”. We therefore adopted the expression “systematic barriers, such as rules and customs” for semantic equivalence. The expression of choices in the original questionnaire (strongly disagree–strongly agree) was considered inadequate because of the variance in natural Japanese response patterns contingent on the item type. We therefore modified the expression of choices for some items inquiring about knowledge or experience (i.e., not at all–very well or none–very much) until we had a final version of LGBT-DOCSS-JP (S1 Appendix). Content validity and cross-cultural validity were established through these processes.

In addition, four scales were used to measure the construct validity of the LGBT-DOCSS-JP in the second phase: ATLG-J6R, J-GTS-R-SF, J-RWA, and J-BIDR. Each scale is already established and validated in Japanese. We did not use all items on all the scales because of the risk that too many items in the questionnaire would reduce the response rate. ATLG-J6R is a revised version of ATLG-J20, the original full-scale containing 20 items. Both of these scales were validated by Horikawa et al. [22]. We used all six items of the ATLG-J6R. J-GTS-R-SF is a short form of J-GTS-R, and both were validated by Mori et al. [23]. We used all seven items of the “Genderism and Transphobia” subscale of J-GTS-R-SF because Bidell used the same subscale in the previous study. J-RWA was validated by Takano et al. [24]. Bidell selected five items from the RWA (the original English version of J-RWA) to verify the construct validity of LGBT-DOCSS, and used a seven-point Likert-type scale instead of the original nine-point scale to reduce participants’ fatigue. Although it is not common practice to pick only a few items and use them in this way, we followed Bidell’s method for comparison with the previous study. We used all items of the J-BIDR, validated by Tani [25]. In summary, of the four scales used, ATLG-J6R and J-BIDR remained unchanged; for J-GTS-R-SF we used only a subscale, and from J-RWA we used only five of the items following Bidell’s method. We have added some further explanations to the manuscript, as follows:

Page 7, line 219

Both the original full scale, consisting of 20 items, and a revised version, consisting of 6 items, have been validated by Horikawa et al. [23].

Page 7, line 235

J-GTS-R-SF, the abbreviated version of J-GTS-R containing 13 items, was also validated by Mori et al. [24]. …… We used all seven items of the “Genderism and Transphobia” subscale of J-GTS-R-SF, following Bidell’s method [5].

Page 8, line 250

We selected five items focusing on sexual morality, homosexuality, sexual preference, gays and lesbians, and premarital sexual intercourse, in line with Bidell’s study [5]. J-RWA uses a nine-point Likert-type scale, ranging from −4 = very strongly disagree to 4 = very strongly agree, but we used the seven-point Likert-type scale from the LGBT-DOCSS-JP (1 = strongly disagree to 7 = strongly agree) to minimize participants’ fatigue with multiple scale changes, following Bidell’s method [5].

#4

Conclusions: Overall the article makes an interesting contribution to the field. However, as the author mentions, the results come from very specific clinics which are not necessarily representative of other care contexts. This limitation should therefore be reflected more clearly in the conclusions. It can only be concluded that the new version of the scale could be relevant for studying the abilities to work with sexually diverse people of Japanese professionals, but a larger investigation will need to confirm these results.

We agree and have added some additional sentences to the final part of the paper, as follows:

Page 25, line 599

In conclusion, LGBT-DOCSS-JP could be relevant for assessing the clinical skills of Japanese healthcare professionals working with people of diverse sexuality and gender. Nevertheless, a more extensive investigation is needed to confirm these results.

Reviewer 2

#1

Overall, the article adds to the current body of research. The creation of the additional factor of Clinical training is very interesting and could be developed further. The researchers did a great job with their analysis and the tools they used to establish the validity and reliability. However, there are some concerns.

#2

The first concern is the researchers did not utilize a professional interpreter to interpret the original English version of the survey.

Your comments about the rigor of the translation process are appreciated. Although we did not hire a professional translator, Dr. Hayashi, who undertook the back-translation, has a track record of back-translating for multiple scale development studies in the past [1-3]. He is a bilingual physician with experience and considerable expertise in developing Japanese-language scales in the medical field.

References

1) Horiguchi R, Mutai R, Gomi M, Sugiyama Y, Iwata H, Satake S, et al. Development of the Adapted Physician Centrality Scale: a cross-sectional study in Japan. Jikeikai Medical Journal. 2021;68(4):89-98.

2) Kaneko M, Okada T, Aoki T, Inoue M, Watanabe T, Kuroki M, et al. Development and validation of a Japanese version of the person-centered primary care measure. BMC Prim Care. 2022;23(1):112. Epub 20220510. doi: 10.1186/s12875-022-01726-7. PubMed PMID: 35538437; PubMed Central PMCID: PMCPMC9088030.

3) Mutai R, Sugiyama Y, Yoshida S, Horiguchi R, Watanabe T, Kaneko M, et al. Development and validation of a Japanese version of the Patient Centred Assessment Method and its user guide: a cross-sectional study. BMJ Open. 2020;10(11):e037282. Epub 20201124. doi: 10.1136/bmjopen-2020-037282. PubMed PMID: 33234616; PubMed Central PMCID: PMCPMC7689105. 

#3

It appears from the article, the researchers used the feedback from the participants to modify words that may not translate well, as explained in lines 334-343. It is important to note, were the participants native Japanese speakers.

Following the reviewer’s suggestion, we have revised this information about the participants, as follows:

Page 5, line 142

After completing a prototype version, we recruited participants who were native Japanese speakers via personal connections by purposive sampling to conduct a cognitive debriefing.

Page 12, line 378

In the first phase, we recruited eight native Japanese-speaking participants,…

#4

Are there different versions of Japanese spoken based on geographic location in Japan? As a reviewer, I have concerns if this is enough to translate a document.

Yes, there is a widely spoken ‘standard’ Japanese and other dialects of Japanese that vary by region. However, almost all Japanese are able to use standard Japanese because primary education in Japan is conducted in standard Japanese, and the penetration rate of primary education is almost 100%. In addition, healthcare professionals – the target population of this scale – have high levels of education, indicating that a scale using standard Japanese would be acceptable and appropriate for them. We added the following sentence to clarify this point:

Page 4, line 121

We used standard Japanese, the version of the Japanese language that is widely used across the country and appropriate for almost all Japanese healthcare professionals to use and understand.

#5

The second concern is the paper does not identify, discuss, or address the cultural experiences of being LGBT in Japan. What is that overall experience and the healthcare disparities associated with being LGBT. Do healthcare disparities exist in Japan among LGBT people? If so, which ones and what role does the healthcare professional play? In Japan, are healthcare providers allowed to turn a patient away? Are there limitations in the care the healthcare provider can provide. Needs some clarification and more in-depth backgrounds.

We appreciate your thoughtful comments on this relevant issue. Yes, healthcare disparities exist in Japan among LGBT people, and we concur that providing more detail on this aspect provides a stronger rationale for the study. Based on your comments, we revised the introduction substantially: 

Page 3, line 76

In Japan, health and healthcare disparities exist for LGBT people. LGBT people have higher rates of psychological distress, suicidal ideation, and suicide attempts than cisgender heterosexuals [2]. Although it is illegal to reject a patient on the basis of sexual orientation or gender identity, access to healthcare is limited and problematic, with about 50% of transgender people having had an unpleasant experience when visiting a medical institution, and about 50% hesitating to visit a medical institution when unwell [8]. Gender-affirming hormone therapy is not covered by national health insurance, and as of April 2023, only eight medical institutions have been authorized to perform gender-affirmation surgery by national health insurance. An anonymous survey of more than 8,000 physicians in Japan revealed that 46% had not seen any LGBT patients in the past 5 years [9]. In a study of nursing directors, more than 30% of hospitals limited ICU visitation and end-of-life care attendance to relatives and opposite-sex partners and did not allow same-sex partners [10]. These problems may partially reflect healthcare professionals’ lack of knowledge and experience with LGBT health issues.

#6

The rest of my review is comprised of recommendations. There is a lack of consistency of the terms throughout the paper, including ‘education’ and ‘care’. It is my recommendation that when mentioning education be specific to health care education. When mentioning care it should be specific to ‘healthcare’ or ‘patientcare’. For example line 50 and lines 70-72 should be clear that it is health care education.

Following your recommendation, we applied the terms ‘healthcare’ and ‘healthcare education’ to clarify the meaning at appropriate points throughout the manuscript.

#7

Line 72 … missing the word patient, not just caring for sexual and gender minorities but patient care. Line 80 Be specific- lack of a suitable LGBT healthcare competence evaluation scale.

We have revised the sentence as follows:

Page 3, line 71

LGBT-DOCSS has been used in recent studies to show that sufficient healthcare education and experience improves medical students’ clinical skills [6] and to develop guidelines and recommendations for the patient care of sexual and gender minorities [7].

Page 4, line 101

One major obstacle is the lack of a suitable LGBT healthcare competence evaluation scale.

#8

Another recommendation is providing the readers with a background/introduction into the medical school. For example, Line 74- the term medical school is specified- is this medical school for medical doctors? Does this include nursing? Try to be more specific if possible. Since most participants were nurses their required education should be addressed and discussed.

We used the term ‘medical school’ to refer to training for medical students, and ‘nursing school’ for nursing students. We addressed medical education and nursing education separately. The focus on nursing education was made more explicit, as follows:

Page 4, line 97

The model core curriculum for nursing education also sets understanding and assessing sexual diversity as a learning objective [13]. However, in a previous study, approximately 80% of nurses indicated that they did not learn about sexual and gender minorities in their basic nursing education [14].

#9

I recommend on Line 358 when providing percentages, they should provide the n so the reader knows that number that percentage is representing.

Following the reviewer’s comment, we revised the manuscript as follows:

Page 13, line 415

Overall, among 381 participants who were analyzed in the first survey, 47.5% (n = 181) of the sample were in their 20s, 69.6% (n = 265) identified as cisgender females, and 78.2% (n = 298) identified as heterosexual. The majority of participants (66.1%, n = 252) were nurses, and 87.3% (220 of 252) of the responding nurses were cisgender females.

#10

Introduction

There is a more updated version to Healthy People, Healthy People 2030.

Thank you for the comment. We have now referred to the 2030 version.

#11

Are there guidelines on how much time should be spent? The introduction should address the culture of Japan towards LGBT patients.

Although not an established guideline, a study in the USA recommended providing 35 hours of education on LGBT issues to medical students. In addition, a study showed the median (IQR) amount of time spent on the education of healthcare for LGBT individuals was 1 hour (0–2) in Japan, compared with 4 hours (2–6) in the USA and Canada. We have now included that detail in the manuscript, as follows:

Page 4, line 95

However, a recent survey showed that medical schools in Japan spend a median of 1 hour (interquartile range (IQR) 0–2 hours) compared with 4 (IQR 2–6) hours in the USA and Canada [12].

#12

LGBT-DOCSS does tell how to teach this content but rather identifies areas of weakness in the healthcare practitioners (line 79).

Thank you for your comment. Yes, LGBT-DOCSS cannot be used as a guideline for education as it stands. Instead, as Bidell suggests, it is a tool that could be used to facilitate trainees’ and healthcare providers’ self-exploration of their LGBT clinical skills, and improve their weaknesses. Educational practice and evaluation are intertwined, and it is our vision that the LGBT-DOCSS will be used as an evaluation of education, leading to identifying areas and domains for further improvement in education. We revised the manuscript, as follows:

Page 4, line 102

Bidell concluded that the LGBT-DOCSS could be used to facilitate trainees’ and healthcare professionals’ self-exploration of their LGBT clinical skills to improve the areas of weakness [5].

#13

First Phase-

The scoring was explained, specifically the reverse scoring of the attitudinal scale.

Why are they so high and Clinical training, clinical preparedness, and basic knowledge are all low, clinical preparedness is really low, especially when looking at age- wouldn’t age have higher scores? They don’t explain the scoring.

We apologize that this information was unclear. In the revised version, we now discuss some possible reasons for the high scores on the attitudinal awareness subscale. (Page 23, line 516 “However, the Attitudinal Awareness subscale score was as high in this study as in the original study (6.30 (0.82) vs. 6.52 (0.72))……A higher score for Attitudinal Awareness is therefore reasonable.”) It is also possible that there was a selection bias and that the scores were higher overall as a result of more LGBT-aware respondents. (Page 25, line 579 “Second, there was some self-selection bias. ……Those who had more awareness of sexual and gender minorities may have been more inclined to respond. Furthermore, the overall response rate was limited despite our reminder emails. It is therefore conceivable that our scale scores were overestimated when compared with the general population.”)

We also discuss a possible reason for older participants’ high scores on the clinical preparedness subscale. (Page 23, line 530 “Older participants scored higher on the Clinical Preparedness subscale, probably because of increased clinical expertise acquired over their years of professional healthcare experience.”)

A further possible reason for the low scores on clinical training, clinical preparedness, and basic knowledge is lack of healthcare education. We have added the following explanation:

Page 23, line 513

The low scores on these three subscales may be because of insufficient healthcare education in Japan.

#14

Results

Close to 70% were cisgender females, are the majority of healthcare workers female in Japan? Is this representative? What role/impact does most participants being nurse and females have? Does this impact the interpretation of the survey?

Thank you for your important remarks. According to national statistics in 2020 [1], 92% of nurses in Japan are female. In this study, 87% (220 of 252) of the responding nurses were cisgender women. Thus, this percentage of female nurses is representative of Japan. We have now added a supplemental table (Table S1: The scores of cisgender participants and each professional healthcare specialization) that shows cisgender women tend to have higher scores, especially attitudinal awareness scores, compared with cisgender men. (Our original Table S1 is now S2). It is possible that the scores for the group as a whole were relatively high. Furthermore, although the different educational and work backgrounds preclude simple comparisons between professional healthcare specializations, the scores for the sample as a whole may have been most heavily influenced by the scores of the nursing participants. Therefore, we recommend caution when, for example, making comparisons in populations with different gender distributions of healthcare workers, as in international comparisons. We revised the manuscript to emphasize this limitation. However, the main purpose of the current study was to validate the scale characteristics for use with Japanese health professionals, and the results showed sufficient reliability and validity for our sample. Further verification of the reliability and validity of this scale in other populations is warranted, as different populations will have different scale characteristics.

Page 13, line 418

The majority of participants (66.1%, n = 252) were nurses, and 87.3% (220 of 252) of the responding nurses were cisgender females. The mean (standard deviation (SD)) LGBT-DOCSS-JP total score was 4.16 (0.74) and the median (IQR) score was 4.11 (3.67–4.72). S1 Table shows the scores of cisgender participants and each professional healthcare specialization. The cisgender female participants’ mean (SD) total score was 4.17 (0.68), and the nurses’ mean (SD) total score was 4.13 (0.68).

Page 25, line 589

Fourth, there was a bias of gender and professional healthcare specialization among the participants. In this study, the percentages of cisgender females and nurses were both close to 70%, with 87.3% of responding nurses being cisgender females. According to the national statistics in 2020, 92% of nurses in Japan are female [41]. Thus, the percentage of female nurses in this study is representative of Japan. However, it is clear that the overall population scores were heavily influenced by cisgender female and nurse participants. Therefore, caution should be exercised when making comparisons with populations that have different distributions of gender and specialty areas among healthcare professionals, as is the case when making international comparisons.

Reference

1) Ministry of Health, Labour and Welfare. Overview of the Year 2020 Health Administration Report (Employment Medical Personnel): Public health nurses, midwives, nurses, and assistant nurses. https://www.mhlw.go.jp/toukei/saikin/hw/eisei/20/dl/kekka1.pdf

#15

Discussion

The researchers state despite the lack of official practical training, many healthcare professionals are aware and provide care- did they collect data on how many of participants provide care to LGBT and how many hours of training they received? 

Thank you for your comment and questions. We did not collect data on how many participants provided care to LGBT people and how many hours of LGBT healthcare training they received. These details are important but were beyond the scope of the study. Our comment on this point was speculation, and we have now rewritten the comment, as follows:

Page 23, line 503

It should be noted that despite the lack of official practical training, many healthcare professionals may be aware of the needs of LGBT patients and provide healthcare for them regularly, increasing their clinical preparedness through trial and error.

#16

Is there continued education or other less formal training and education settings offered? Sentences 431-436- who is responsible for the education of the nurses? In one sentence the authors are discussing school and the next sentence the authors are talking about continuing education.

There needs to be clarity because you cannot compare the two to each other. For example, “line 431 …over 90% of hospitals in Japan do not provide nursing training with LGBT content, despite the demand for education… this is much worse than the 28% of baccalaureate nursing programs’- these two are not comparable.

To our knowledge, there is no public system in Japan to provide continuing education on LGBT issues for healthcare professionals. A few limited healthcare institutions may offer education. Generally, the director of nursing is responsible for the education of hospital nurses. More than 90% of respondents in a Japanese survey of nursing directors indicated that they do not provide continuing nursing education about LGBT patients, and 80% of nurses reported that they did not learn about LGBT issues in their basic nursing education. Following the reviewer’s comment, we revised the manuscript as follows:

Page 22, line 493

Approximately 80% of nurses indicated that they did not learn about sexual and gender minorities in their basic nursing education in Japanese nursing schools [14] – a much higher proportion than the 28% of baccalaureate nursing programs in the USA that did not teach LGBT sexual health at all [37]. In a survey of nursing directors of 252 hospitals in Japan, more than 90% indicated that they do not provide continuing nursing education on LGBT issues, and approximately 10% reported that healthcare education is not necessary [10].

#17

You state however, many healthcare professionals’ are aware of LGBT needs and treat them regularly- what is this based on? Did you measure how many patients the HCW sees?

Thank you for raising this point. Unfortunately we were not able to measure how many patients the healthcare workers treated. Because this is a matter of speculation, we have rewritten that sentence, as follows:

Page 23, line 503

It should be noted that despite the lack of official practical training, many healthcare professionals may be aware of the needs of LGBT patients and provide healthcare for them regularly, increasing their clinical preparedness through trial and error.

#18

Line 394 Why would the scores for clinical preparedness be higher among older age groups? 

Although a matter of conjecture, it may be that the longer one has been in the medical profession, the more experience one has with a variety of patients, and thus, the higher one’s clinical preparedness may be with LGBT patients. We discuss this further as follows:

Page 23, line 530

Older participants scored higher on the Clinical Preparedness subscale, probably because of increased clinical expertise acquired over their years of professional healthcare experience. This is inconsistent with a previous study of healthcare providers in the USA, which reported no significant difference in LGBT-DOCSS scores based on age [39]. Further research in other settings is required to confirm these results.

#19

Line 436 What is meant by school policy? Is there a governing body over the requirements of education or is it done on a school to school/institutional to institutional basis? 

We apologize for using confusing terminology, which we have now revised in the updated manuscript. We meant each school’s policy of education. Although the model core curriculum was formulated [1-2], the content and extent of education can vary by school and curriculum. Yamazaki et al. concluded that some faculties and schools lack preparation and knowledge to implement LGBT education, including a lack of suitable instructors and formal school policies [3]. We revised this part, as follows:

Page 22, line 499

These training issues in Japan are mainly due to the unavailability of suitable instructors and most schools lacking policies regarding LGBT healthcare education [12, 37, 38].

References

1) Medical Education Model Core Curriculum Coordination Committee; Medical Education Model Core Curriculum Expert Research Committee. Model Core Curriculum for Medical Education in Japan, AY 2016 Revision 2017 [cited 11 July 2023]. Available from: https://www.mext.go.jp/component/a_menu/education/detail/__icsFiles/afieldfile/2018/06/18/1325989_30.pdf.

2) Committee for Fostering Human Resources in Nursing Education. Model Core Curriculum for Nursing Education in Japan: Learning objectives aiming for acquiring “core abilities of nursing practice in bachelor’s degree program” 2017 [cited 6 November 2023]. Available from: https://www.mext.go.jp/content/20200428-mxt_igaku1217788_4.pdf.

3) Yamazaki Y, Aoki A, Otaki J. Prevalence and curriculum of sexual and gender minority education in Japanese medical school and future direction. Med Educ Online. 2020;25(1):1710895. doi: 10.1080/10872981.2019.1710895. PubMed PMID: 31931679; PubMed Central PMCID: PMCPMC7006669.

#20

450 What is meant by same-sex partnership schemes.

Line 450 ‘where a succession of same-sex partnership schemes have been instituted at the municipal level since 2015, and the proportion of citizens. What does this mean? 

Japan is the only G7 country that has not introduced same-sex marriage or partnership, which guarantees the same rights as marriage, at the national level. However, the 2010s saw a growing movement among citizens to recognize same-sex partnerships at the municipal level. Currently, more than 300 municipalities recognize same-sex partnerships, with a population coverage rate of more than 70%. In contrast, government policy has not kept pace with public opinion and municipal policy, and no legislation has yet been passed. To clarify the context and what we mean by same-sex partnership schemes, we revised the manuscript, as follows:

Page 23, line 517

These findings align with the social milieu of Japan. Japan is the only G7 country that has not introduced same-sex marriage or partnership that guarantees the same rights as marriage, at the national level. However, the 2010s saw a growing movement among citizens to recognize same-sex partnerships at the municipal level. Currently, more than 300 municipalities recognize same-sex partnerships, with a population coverage rate of more than 70%, and the proportion of citizens showing favorable views of LGBT individuals is increasing [21]. In contrast, government policy has not kept pace with public opinion and local government policy, and no legislation has yet been passed.

---

## [Editor Report · Decision Letter 1]

29 Jan 2024

Development and validation of the Japanese version of the Lesbian, Gay, Bisexual, and Transgender Development of Clinical Skills Scale

PONE-D-23-21890R1

Dear Dr. Kanakubo,

We’re pleased to inform you that your manuscript has been judged scientifically suitable for publication and will be formally accepted for publication once it meets all outstanding technical requirements.

Kind regards,

Daniel Demant, PhD, MPH, GradCertHEd, BAppSocSc

Academic Editor

PLOS ONE

Additional Editor Comments (optional):

The authors have addressed all comments sufficiently and, in many cases, beyond what would have been expected. The revised manuscript fulfils all requirements for publications.
---

## [Editor Report · Acceptance letter]

4 Mar 2024

PONE-D-23-21890R1 

PLOS ONE

Dear Dr. Kanakubo, 

I'm pleased to inform you that your manuscript has been deemed suitable for publication in PLOS ONE. Congratulations! Your manuscript is now being handed over to our production team.

Kind regards, 

on behalf of

Dr. Daniel Demant 

Academic Editor

PLOS ONE